# Evidence of factors influencing delays in the diagnosis and treatment of bipolar disorder in adolescents and young adults. Protocol for a systematic scoping review

**Kamyar Keramatian**[1,2,3]*, **Emma Morton**[4], **Alexander Levit**[1], **John-Jose Nunez**[1,3]

1 Department of Psychiatry, University of British Columbia, Wesbrook Mall, Vancouver, BC, Canada,
2 Coastal Early Psychosis Intervention Program, North Vancouver, British Columbia, 3 Mood Disorders Clinic, Djjawad Movafighan Centre for Brain Health, Vancouver, British Columbia, 4 School of Psychological Sciences, Monash University, Monash, Australia

* kamyar.keramatian@vch.ca

**Data Availability Statement:** No datasets were generated or analyzed during the current study. All

## Abstract

### Background

Bipolar Disorder (BD) is a complex psychiatric condition that typically manifests during late adolescence and early adulthood. Over the past two decades, international studies have reported that BD often goes unrecognized and untreated for several years, which can lead to negative clinical and functional outcomes. However, the components of delay in the diagnosis and treatment of BD and various factors influencing those components have not been systematically explored.

### Objectives

The scoping review described in this protocol aims to map the existing literature on potential factors that influence delays in the treatment of BD in adolescents and young adults, in order to identify the knowledge gaps and future research and policy priorities.

### Methods

This protocol for a systematic scoping review will be reported using the Preferred Reporting Items for Systematic Reviews and Meta-Analyses guideline (PRISMA-ScR). We will search the electronic databases of MEDLINE (OVID), EMBASE, PsycINFO and CINAHL for peer-reviewed primary research articles published in academic journals. Grey literature will not be explored due to resource limitations. A conceptual framework based on the *Model of Pathways to Treatment* by Scott and colleagues was used as a foundation for our search and extraction strategy to ensure all components of delay and potential factors influencing each component are explored. Two independent reviewers will screen the references retrieved by the literature search and select relevant studies based on our inclusion criteria. The data from included studies will be synthesized into a narrative summary, and implications for future research, practice and policy will be discussed.

relevant data from this study will be made available upon study completion.

**Funding:** The author(s) received no specific funding for this work.

**Competing interests:** Kamyar Keramatian has served on the scientific advisory board of AbbVie. John-Jose Nunez has received research funding from an unrestricted research grant from Pfizer Canada. Emma Morton and Alexander Levit have no competing interests to declare. This does not alter our adherence to PLOS ONE policies on sharing data and materials.

## Discussion

To the best of our knowledge, this will be the first scoping review to explore the potential factors that influence delays in the treatment of BD in adolescents and young adults. We intend to disseminate the review results through academic conferences and publication in a peer-reviewed journal.

## Introduction

Bipolar Disorder (BD) is a complex and severe psychiatric condition characterized by recurrent episodes of depression and mania or hypomania. The median age at onset of BD has been estimated to be 17.5 years [1], suggesting that in at least half of individuals with BD the symptoms first emerge before adulthood. Data from the World Health Organization Global Burden of Disease study ranked BD the 4th leading cause of disability worldwide among the 10–24 year age group [2]. Despite its relatively high prevalence and large disability burden, BD often goes unrecognized and untreated for several years. In one of the earliest published reports on this subject, Hirschfeld and colleagues published the results of an American comparative study that assessed the experiences of individuals with BD in years 1992 and 2000 [3]. In 2000, about one-third of individuals with BD reported that they waited over 10 years after the onset of symptoms before seeking professional help. Additionally, 69% of individuals with BD reported having been initially misdiagnosed, with the most common inaccurate diagnosis being unipolar depression (60%), followed by anxiety disorders (26%), schizophrenia (18%), personality disorders (17%) and substance-related disorders (14%). Those who were initially misdiagnosed had received 3.5 other diagnoses and consulted 4 physicians before receiving a diagnosis of BD [3]. These findings were comparable to the ones reported in 1992, suggesting a lack of improvement in early identification and early intervention for almost a decade. Findings from other countries including Japan [4], Nepal [5], India [6], Singapore[7], China [8], Bangladesh [9], Ethiopia [10], Australia [11] also showed significant delay between the onset of symptoms until receiving the diagnosis and initiating appropriate treatment.

Delays in diagnosis and treatment tend to be even more prolonged among those with paediatric-onset BD. In an American multisite retrospective study, childhood-onset, adolescent-onset and early adult-onset BDs were associated with 16.8, 11.5 and 4.6 years treatment delays respectively [12]. More recently, data from a Canadian multicenter naturalistic study suggested the median delay of 15 years (IQR: 8.2–25.5) for pediatric-onset BD (<18 years) and 5.0 years (IQR: 1.0–12.0) for adult-onset BD [13]. Delays are longer for pediatric-onset BD despite this onset being associated with a more serious course [14].

Prolonged delays in the diagnosis and subsequent treatment of BD have been shown to have serious consequences for those with pediatric-onset BD, and more broadly in the BD population. Impacts include disruption of crucial age-specific developmental tasks [15], greater severity and frequency of mood episodes [16], higher number of hospitalizations [17], higher number of comorbidities [18,19], and elevated risk of suicide [18,20,21]. In addition, delay in the diagnosis of BD has been shown to be associated with significantly higher healthcare costs [22] as well as higher indirect costs due to work loss [23,24]. Given the more severe course and especially prolonged delay in diagnosis of BD in youth, there is an especially urgent need to better understand factors contributing to the delay in the diagnosis and treatment of BD–to inform targeted early identification strategies.

This scoping review aims to identify factors involved in the delayed diagnosis and treatment of BD in adolescents and young adults. We operationally defined delayed treatment as the time between the onset of BD symptoms and initiation of best practice evidence-based interventions. Given the complex nature of delayed treatment and in order to ensure all key contributing factors are being explored, we used the *Model of Pathways to Treatment* by Scott and colleagues [25] to guide our search strategy. According to this multidimensional framework, the total time from the onset of mood symptoms until treatment initiation is divided into four sequential intervals. **The Appraisal interval** is defined as the time from the onset of mood symptoms to perceiving a reason to discuss such symptoms with a healthcare professional (HCP). **The Help-seeking interval** describes the time from perceiving a reason to discuss mood symptoms with a HCP to the first consultation regarding those symptoms. **The Diagnostic interval** represents the time between the first appointment with a HCP and receiving the accurate diagnosis of BD. **The Pre-treatment interval** describes the time between accurate diagnosis of BD and initiation of effective, evidence-based interventions. Subsequently, potential factors that may influence the duration of each interval are identified. These potential contributing factors are grouped into four interconnected categories. **Patient factors** include individual characteristics as well as family, social and cultural factors. **Disease factors** refer to clinical aspects of BD (e.g., polarity of the first episode, type and severity of symptoms). **Healthcare provider and mental health system factors** include mental health service availability, accessibility, and affordability as well as therapeutic alliance, diagnostic accuracy, and implementation of evidence-based practice.

## Methods

A scoping review was deemed as the most suitable method to map the existing literature on potential factors that influence delays in the treatment of BD in adolescents and young adults. In general, a scoping review provides an overview of a broad topic and is a preferable approach when the main purpose of our review is to identify key factors related to a concept or to identify and analyse gaps in the knowledge base [26]. This protocol has been registered with Open Science Framework (https://osf.io/qcug7).

We will conduct our scoping review in accordance with the methodological framework outlined by Arksey and O'Malley [27], and will report using the Preferred Reporting Items for Systematic Reviews and Meta-Analyses guideline (PRISMA-ScR) [28]. Arksey and O'Malley's methodological described the following 6 stages in conducting a scoping review: 1) identification of the research question, 2) identification of relevant studies, 3) selection of studies, 4) charting the data, 5) collating, summarizing and reporting of results, and 6) consultation with stakeholders (optional). We will also incorporate recommendations by Levac and colleagues [29] to enhance each stage of the Arksey and O'Malley methodological framework.

### Stage 1: Identifying the research question

Before arriving at the final research question, we identified the main concept of delayed diagnosis and treatment of BD in adolescents and young adults. We then used the *Model of Pathways to Treatment* [25] as a roadmap to explore the components of delay in the diagnosis and treatment of youth with BD and various factors that can influence the time taken for each component and to select our search terms. Our primary research question is "What is known from the existing literature about patient, disease and healthcare system-provider factors that influence delay in the diagnosis and treatment of BD in adolescents and young adults?" In this work, we define adolescence and young adulthood as those experiencing the onset of BD between the ages of 13 and 24, centered upon the mean age of diagnosis, 18 [DSM5]. This

range starts with the typical age used as the transition from pre-puberty topuberty [30], and coincides with an age of increasing incidence of mood disorders as well as substance use disorders and schizophrenia spectrum disorders [31]. The range ends with a common age considered to be the transition from young to middle adulthood as it is around this age that a majority of adults have started to form a family by cohabitation, marriage, or becoming a parent [32] and also coincides with complete maturation of brain regions & neurocircuitry involved in executive and emotional functions [33]. This is also a range similar to that used by early intervention services [34,35].

## Stage 2: Identification of relevant studies

We will use the Population-Concept-Context (PCC) framework [36] to determine our eligibility criteria as outlined in Table 1. The following electronic databases will be searched: MEDLINE (OVID), EMBASE (OVID), PsycINFO (EBSCOHost), and CINAHL (EBSCOHost).

## Stage 3: Search strategy and study selection

*Search strategy*: We used the *Model of Pathways to Treatment* by Scott and colleagues [25] as a foundation for our search strategy. Using this framework, a list of potential factors that could influence each component of delay in the treatment of BD were identified. We then performed an informal literature search to identify a preliminary list of search terms related to the list of the potential contributing factors. The final search strategy will be was iteratively developed with support from subject librarians at the University of British Columbia (Table 2).

*Study selection*: Once the final search strategy is determined, two independent reviewers will first screen the tiles and abstracts of the references retrieved by the literature search, using Covidence, which is a web-based collaborative software platform for conducting systematic reviews. The full texts of the relevant articles will then be reviewed by the same reviewers who will decide on the inclusion of the articles based on our inclusion criteria. Any disagreement between the two reviewers will be resolved by consensus or a third reviewer from the team.

## Stage 4: Charting the data

The protocol authors will develop a tabular chart to extract the relevant information from selected articles. A preliminary list of appropriate variables according to the Population-Concept-Context (PCC) framework can be found in Table 3.

**Table 1. Eligibility criteria according to Population-Concept-Context (PCC) framework.**

| | |
|---|---|
| Population | Adolescents and young adults with onset of mood symptoms or study enrollment at a mean age of 13 to 24 years, with diagnoses of bipolar spectrum disorder (including bipolar I disorder, bipolar II disorder, and bipolar not-otherwise specified, or unspecified/other specified bipolar and related disorder. Studies that include mixed populations whereby more than 80% of participants have bipolar spectrum disorder or where bipolar disorder specific data can be extracted will also be included. |
| Concept | Patient, disease and healthcare system-provider factors related to the components of delay in the diagnosis and treatment of bipolar spectrum disorder. |
| Context | Clinical setting: All settings including inpatient, outpatient, primary care and specialized. Geography: No limits Publication type: Primary qualitative and quantitative research published in peer-reviewed journals. Language: English. Publication date: 2000–2023. |

**Table 2. Preliminary list of search terms.**

| OVID-MEDLINE (R) | |
|---|---|
| (and Epub Ahead of Print, In-Process, In-Data-Review & Other Non-Indexed Citations, Daily and Versions) | |
| **Illness**<br>Terms: 6 | bipolar disorder/ OR "bipolar disorder" OR mania OR manic OR hypomania OR hypomanic |
| AND | |
| **Age group**<br>Terms: 10 | Adolescent/ OR adolescen* OR "Young Adult/" OR "young adult*" OR Pediatrics/ OR pediatric* OR paediatric* OR youth* OR teen* OR juvenile* |
| AND | |
| **Components of delay**<br>Terms: 60 | "duration of untreated" OR delayed diagnosis/ OR "delayed diagnosis" OR "delay in treatment" OR "treatment latency" OR "latency to treatment" OR "symptom overlap" OR "symptom recognition" OR "symptom appraisal" OR "symptom attribution" OR "symptom misattribution" OR "symptom perception" OR "symptom awareness" OR belief* OR attitude* OR heuristic* OR "cognitive bias*" OR "coping style*" OR "self regulation" OR "screening" OR "help seeking" OR "information seeking" OR Help seeking behavior/ OR Health literacy/ OR Self-efficacy/ OR "Self-efficacy" OR "self medication" OR "self treatment" OR stigma* OR Health services accessibility/ OR "access to care" OR "access to treatment" OR "access to medication*" OR affordab* OR "waiting time" OR polarity OR underreporting OR under reporting OR "therapeutic alliance" OR clinical competence/ OR "clinical competenc*" OR diagnostic error/ OR "diagnostic error*" OR "diagnostic valid*" OR "diagnostic reliability" OR "diagnostic criter*" OR "diagnostic accuracy" OR "stability of diagnosis" OR "diagnostic stability" OR misdiagnosis OR "Referral and Consultation"/ OR referral* OR consult* OR insight OR Attitude to Health/ OR Patient Compliance/ OR adherence OR compliance OR implementation OR policy |
| OVID-MEDBASE | |
| **Illness**<br>Terms: 6 | bipolar disorder/ OR "bipolar disorder" OR mania OR manic OR hypomania OR hypomanic |
| AND | |
| **Age group**<br>Terms: 10 | Adolescent/ OR adolescen* OR "Young Adult/" OR "young adult*" ORPediatrics/ OR pediatric* OR paediatric* OR youth* OR teen* OR juvenile* |
| AND | |
| **Components of delay**<br>Terms: 60 | "duration of untreated" OR "delayed diagnosis" OR delayed diagnosis/ OR "delay in treatment" OR "treatment latency" OR "latency to treatment" OR "symptom overlap" OR "symptom recognition" OR "symptom appraisal" OR "symptom attribution" OR "symptom misattribution" OR "symptom perception" OR "symptom awareness" OR belief* OR attitude* OR heuristic* OR "cognitive bias*" OR "coping style*" OR "self regulation" OR screening OR "help seeking" OR "information seeking" OR Help seeking behavior/ OR Health literacy/ OR "Self-efficacy" OR "self medication" OR "self treatment" OR stigma* OR Health Care Access/ OR "access to care" OR "access to treatment" OR "access to medication*" OR affordab* OR "waiting time" OR polarity OR underreporting OR "under reporting" OR "therapeutic alliance" OR clinical competence/ OR "clinical competenc*" OR diagnostic error/ OR "diagnostic error*" OR "diagnostic valid*" OR "diagnostic reliability" OR "diagnostic criter*" OR "diagnostic accuracy" OR "stability of diagnosis" OR "diagnostic stability" OR misdiagnosis OR Patient Referral/ OR Referral* OR Consult* OR insight OR Attitude to Health/ OR Patient Compliance/ OR adherence OR compliance OR implementation OR policy |
| EBSCOHost-PsycINFO | |
| **Illness**<br>Terms: 6 | DE "Bipolar Disorder" OR "bipolar disorder" OR mania OR manic OR hypomania OR hypomanic |
| AND | |
| **Age group**<br>Terms: 10 | DE "Adolescent Health" OR adolescen* OR DE "Emerging Adulthood" OR young adult* OR DE "Pediatrics" OR pediatric* OR paediatric* or youth* or teen* or juvenile* |
| AND | |

*(Continued)*

**Table 2.** (Continued)

| | |
|---|---|
| **Components of delay**<br>Terms: 60 | "duration of untreated" OR "delayed diagnosis" OR "diagnostic delay" OR "delay in treatment" OR "treatment latency" OR "latency to treatment" OR "symptom overlap" OR "symptom recognition" OR "symptom appraisal" OR "symptom attribution" OR "symptom misattribution" OR "symptom perception" OR "symptom awareness" OR belief* OR attitude* OR heuristic* OR "cognitive bias*" OR "coping style*" OR "self regulation" OR screening OR "help seeking " OR "information seeking" OR DE "Help Seeking Behavior" OR DE "Health Literacy" OR DE Self-Efficacy OR "Self-efficacy" OR "self medication" OR "self treatment" OR stigma OR DE Health Care Access OR "access to care" OR "access to treatment" OR "access to medication" OR affordab* OR "waiting time" OR polarity OR underreporting OR "under reporting" OR "therapeutic alliance" OR "clinical competenc*" OR "diagnostic error*" OR "diagnostic valid*" OR "diagnostic reliability" OR "diagnostic criter*" OR "diagnostic accuracy" OR "stability of diagnosis" OR "diagnostic stability" OR misdiagnosis OR DE Self Referral OR DE Patient Referral OR "Referral*" OR "Consult*" OR insight OR DE "Health Attitudes" OR "attitude to health" OR DE Treatment Compliance OR adherence OR compliance OR implementation OR policy |

| | EBSCOHost-CINAHL |
|---|---|
| **Illness**<br>Terms: 6 | MH "Bipolar Disorder" OR bipolar disorder OR mania OR manic OR hypomania OR hypomanic |
| | AND |
| **Age group**<br>Terms: 10 | MH "Adolescent Health" OR adolescen* OR MH "Emerging Adulthood" OR young adult* OR MH "Pediatrics" OR pediatric* OR paediatric* OR youth* OR teen* OR juvenile* |
| | AND |
| **Components of delay**<br>Terms: 59 | "duration of untreated" OR "delayed diagnosis" OR "diagnostic delay" OR "delay in treatment" OR "treatment latency " OR "latency to treatment" OR "symptom overlap" OR "symptom recognition" OR "symptom appraisal" OR "symptom attribution" OR "symptom misattribution" OR "symptom perception" OR "symptom awareness" OR belief* OR attitude* OR heuristic* OR "cognitive bias*" OR "coping style*" OR "self regulation" OR screening OR "help seeking " OR "information seeking" OR MH "Help Seeking Behavior" OR MH "Health Literacy" OR MH Self-efficacy OR "Self-efficacy" OR "self medication" OR "self treatment" OR stigma OR MH Health Care Access OR "access to care" OR "access to treatment" OR "access to medication" OR affordab* OR "waiting time" OR polarity OR underreporting OR "under reporting" OR "therapeutic alliance" OR "clinical competenc*" OR "diagnostic error*" OR "diagnostic valid*" OR "diagnostic reliability" OR "diagnostic criter*" OR "diagnostic accuracy" OR "stability of diagnosis" OR "diagnostic stability" OR misdiagnosis OR (MH "Referral and Consultation") OR referral* OR consult* OR insight OR "MH "Attitude to Health" " OR "attitude to health" OR MH "Medication Compliance" OR adherence OR "compliance" OR implementation OR policy |

Forward slashes indicate MeSH Terms in MEDLINE and EMBASE searches; DE and MH indicates subject headings in PsycINFO and CINAHL, respectively. Differences in search term counts reflect differences in MeSH/subject headings available to respective databases.

## Stage 5: Collating, summarizing and reporting the results

In line with recommendations from Levac et al [29], Arksey and O'Malley [27], and Braun and Clarke [37], we will perform stage 5 in three steps:

Step 1. Collating and summarizing the results: We will synthesize the results into a narrative summary to discuss how the findings relate to our research question.

Step 2. Reporting the results: Results will be organized into study aims, methodological designs, major findings and gaps in the literature.

Step 3. Research implications for future research, practice and policy: By understanding factors involved in the delayed diagnosis and treatment of youth with BD and identifying gaps within the literature, this scoping review will inform clinical practice, future research and policy changes.

**Table 3. Preliminary table of charting elements and associated questions for data.**

| General information | Population | Context | Concept |
|---|---|---|---|
| Title | Demographics (age, sex, ethnicity, socioeconomical) | Country | Which component(s) of delay were studied |
| Authors | Inclusion and exclusion criteria | Community (urban/rural/mixed) | Method of calculating the component of the delay |
| Year of publication | Sample size | Clinical setting (inpatient/outpatient; primary care/specialized) | What contributing factors were investigated |
| Publication type | Comorbidities | Operational definition of the diagnosis of bipolar disorder | Method of measuring/determining the contributing factors |
| Study aims | | | What were the impacts of the contributing factors on the component(s) of delay |
| Study design | | | |
| Recruitment procedures | | | |

Overall, the findings will be reported in accordance with the Preferred Reporting Items for Systematic Reviews and Meta-Analyses extension for Scoping Reviews checklist.

## Stage 6: Consultation with stakeholders

As recommended by Levac et al (2010), we will consult with various stakeholders to add to the methodological rigor of the scoping protocol [29]. We will engage with local experts including clinicians, academics as well as individuals with lived experience with BD through the CREST. BD network [38] to validate summarization and interpretation of our work, and to inform the future work we call for.

## Discussion

BD typically manifests during late adolescence and early adulthood. It is now well recognized that BD often goes unrecognized and untreated for several years, which can lead to negative clinical and functional outcomes as well as unnecessary suffering for affected individuals and their families. However, the components of such delays and contributing patient, disease, and healthcare provider/mental health system factors have not been systematically explored yet. To the best of our knowledge, this systematic scoping review will be the first to map the existing literature on potential factors that influence delays in the diagnosis and treatment of BD in adolescents and young adults.

We believe our focus on those with onset in adolescence and young adulthood will help focus the finding of this work. There is ongoing research into the phenomenological distinction around pre-pubertal-onset BD [39].. Similarly, BD onsetting later in life may have distinct biologic underpinning, including neuroinflammation, cerebrovascular injury, and neurodegenerative disease [40,41], and would be occurring in a different psychosocial environment than experienced by youth. Focusing our work on onset in adolescence and young adulthood captures the peak ages of onset and will help determine causes of delay in a population that has long delays and a severe course [42]. For this group, developmental stages, involvement of family and location of treatment may have impacts that are not shared with these other ages. However, we believe this will allow findings from this scoping review to best help identify the knowledge gaps in the literature and inform clinical practice, as well as future research and policy changes. Services in many settings are structured to specifically address pediatric vs youth vs adult MH, so examining delays specific to this group may better inform targeted

interventions at a structural level to improve timely diagnosis and treatment. In addition, we can support recommendations and health promotion for contextual factors relatively unique to this group: families, secondary and tertiary education settings. Investigating delays in this population will also lay a foundation for future work to investigate delays in those experiencing onset in other age groups, and for investigating delays in other psychiatric disorders.

Potential limitations include the exclusion of studies not published in English, secondary research, grey literature, and the focus of this work solely on individuals with onset in adolescence and young adulthood. We excluded secondary studies to avoid redundancy with our own extraction and to ensure our inclusion and exclusion criteria are strictly met, though we acknowledge that findings from these reviews may be relevant. We excluded grey literature to help balance the feasibility of this review with breadth and comprehensiveness [43] though we acknowledge that this may miss some relevant evidence. This work's focus on those with bipolar onset in childhood and adolescence does exclude individuals with onset at other points in life. As previously discussed, we believe this will help focus on scope of this work, but it may also exclude important causes for delay found in studies looking at general bipolar populations, and may exclude causes of delay better described for other age-of-onset, but which may still apply to this age group. A limitation of using the Model of Pathways to Treatment framework is its specificity may mean we will not find relevant studies addressing delays in all of its intervals. However, we believe such a finding would itself be informative, as it would motivate future work to investigate these identified gaps in the literatures. This scoping review will suggest important areas for questioning, and could also be a starting place for consultation/prioritization activities with stakeholders.

## Supporting information

**S1 Checklist. PRISMA-P 2015 checklist.**
(DOCX)

**S2 Checklist. Preferred reporting items for systematic reviews and meta-analyses extension for scoping reviews (PRISMA-ScR) checklist.**
(DOCX)

## Acknowledgments

The authors are grateful for the University of British Columbia librarians, Susan Paterson, BA MLIS, and Dean Giustini, MLS, Med, for their expertise and contribution in the refinement of our search strategy. We are also especially thankful to Rajakumari Pampa Reddy, PhD, for contributing to the earlier stage of this project and participating in the discussions.

## Author Contributions

**Conceptualization:** Kamyar Keramatian, Emma Morton.

**Methodology:** Kamyar Keramatian, Emma Morton, Alexander Levit, John-Jose Nunez.

**Project administration:** Kamyar Keramatian.

**Visualization:** Kamyar Keramatian, Alexander Levit, John-Jose Nunez.

**Writing – original draft:** Kamyar Keramatian.

**Writing – review & editing:** Emma Morton, Alexander Levit, John-Jose Nunez.

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
