## [Decision Letter · Decision Letter 0]

20 Jul 2023

PONE-D-23-11500Evidence of factors influencing delays in the diagnosis and treatment of bipolar disorder in adolescents and young adults. Protocol for a systematic scoping review.PLOS ONE

Dear Dr. Keramatian,

Thank you for submitting your manuscript to PLOS ONE. After careful consideration, we feel that it has merit but does not fully meet PLOS ONE’s publication criteria as it currently stands. Therefore, we invite you to submit a revised version of the manuscript that addresses the points raised during the review process.

We look forward to receiving your revised manuscript.

Kind regards,

Jianhong Zhou

Staff Editor

PLOS ONE

Journal Requirements:

Kamyar Keramatian has served on the scientific advisory board of AbbVie. Emma Morton and Alexander Levit have no competing interests to declare.

3. We noted in your submission details that a portion of your manuscript may have been presented or published elsewhere. Please clarify whether this publication was peer-reviewed and formally published. If this work was previously peer-reviewed and published, in the cover letter please provide the reason that this work does not constitute dual publication and should be included in the current manuscript.

Reviewers' comments:

Reviewer's Responses to Questions

**Comments to the Author**

1. Does the manuscript provide a valid rationale for the proposed study, with clearly identified and justified research questions?

Reviewer #1: Yes

Reviewer #2: Partly

2. Is the protocol technically sound and planned in a manner that will lead to a meaningful outcome and allow testing the stated hypotheses?

Reviewer #1: Yes

Reviewer #2: Partly

3. Is the methodology feasible and described in sufficient detail to allow the work to be replicable?

Reviewer #1: Yes

Reviewer #2: Yes

4. Have the authors described where all data underlying the findings will be made available when the study is complete?

Reviewer #1: Yes

Reviewer #2: Yes

5. Is the manuscript presented in an intelligible fashion and written in standard English?

Reviewer #1: Yes

Reviewer #2: Yes

6. Review Comments to the Author

You may also provide optional suggestions and comments to authors that they might find helpful in planning their study.

Reviewer #1: ABSTRACT

Change subsection of introduction to "background" and "objectives"

Well-written abstract to generate interest in readers

INTRODUCTION

Well-researched and written content based on recent literature review

METHODS

Authors followed PRISMA guidelines and appropriately conducted literature search based on criteria mentioned

DISCUSSION

Authors comply with goals of study and to stem future guidelines for conferences focused on bipolar

Reviewer #2: Thank you for the opportunity to review this paper. The issue of delay in the diagnosis and treatment of bipolar disorder is highly relevant with far-reaching clinical implications. Thus, I believe there is scientific importance to this review topic. I have some feedback regarding the paper's scope, which I describe below.

Overall comments

My primary concern with this review is that the issue of delay in the diagnosis and treatment of bipolar disorder is not specific to the 15-24 age range that you define in your protocol but, instead, a problem that cuts across age groups. Many of the studies that you cite also support that this is not an issue specific to adolescents and young adults. The rationale is not clear enough, based on your cited literature and the existing literature in bipolar disorder more generally, for focusing on a young adult and adolescent population in this review (versus a lifespan focus). References 12 and 13 suggest there is a greater treatment delay for younger individuals but this does not detract from the relevance of this topic for those who are older. I think this paper would have greater value if you were to not restrict to adolescents and young adults but, rather, focus this review on the issue of delay in the diagnosis and treatment of bipolar disorder across the lifespan.

Related to the above comment, how was the age range of 15-24 decided upon? The cited literature describing age-related delays in treatment of BD does not add clarity here (ie, reference 13 compares < 18 to 18+ - your selected population is represented on both sides of this bracket)

The information you aim to gather via the Model of Pathways to Treatment is highly specific, and there is a possibility that the research studies you find through your review may not cover all of the questions you seek to answer. How will you handle the limitation of partial information gathered from different studies (ie, a study covering information on some but not all intervals in the Model) and how will the possibility of incomplete information on these intervals within individual studies impact the utility of your narrative summary?

Specific line item comments

Lines 72-81: The first sentence suggests that the topic of the paragraph covers delays in the diagnosis and treatment of BD among those with pediatric-onset BD but the references here also cover adult onset BD, or cover delays in the diagnosis and treatment of BD more generally. I would recommend re-framing the focus of this paragraph or focusing the paragraph more clearly on your selected population.

Lines 82-89: Similarly, this paragraph also describes consequences associated with delays in diagnosis and treatment of BD, but upon inspection these papers are not solely in pediatric onset BD but across adult populations of BD. I think the last sentence, "Therefore, there is an urgent need to

88 better understand factors contributing to the delay in the diagnosis and treatment of BD –

89 especially in youth – to inform targeted early identification strategies" is not clearly supported by this paragraph.

Table 1: Eligibility criteria. What was the process for determining age cutoffs (15 to 24)?

Lines 179-183: Please add more clarification on the process of consulting with stakeholders: specifically, how you will use this consultation to add methodological rigor and how you will apply the PLA framework for this review.

Lines 192-193: Please add greater explanation on your limitations, specifically the second and third limitations (secondary literature and grey areas).

7. PLOS authors have the option to publish the peer review history of their article (what does this mean?). If published, this will include your full peer review and any attached files.

Reviewer #1: **Yes: **Rikinkumar S. Patel

Reviewer #2: No

---

## [Author Response · Author response to Decision Letter 0]

31 Aug 2023

Dear Dr. Zhou, 

Thank you for your consideration of our manuscript titled “Evidence of factors influencing delays in the diagnosis and treatment of bipolar disorder in adolescents and young adults. Protocol for a systematic scoping review.” We appreciate the opportunity to resubmit the manuscript. We have fully addressed the reviews below, and believe our paper is strengthened from it. 

We have included the original review comments below, enumerated, with our responses in bullet points. We have also included excerpts of all modified text relevant to each response.

Thank you again for your consideration and this opportunity. 

Sincerely, 

Kamyar Keramatian, on behalf of the co-authors

Reviewer #1: ABSTRACT

Change subsection of introduction to "background" and "objectives"

Well-written abstract to generate interest in readers

• Thank you reviewer. We split the former introduction point, into separate Background and Objectives. 

INTRODUCTION

Well-researched and written content based on recent literature review

METHODS

Authors followed PRISMA guidelines and appropriately conducted literature search based on criteria mentioned

DISCUSSION

Authors comply with goals of study and to stem future guidelines for conferences focused on bipolar

• Dear reviewer, thank you for your time and review of your study. We appreciate your positive feedback. 

Reviewer #2: Thank you for the opportunity to review this paper. The issue of delay in the diagnosis and treatment of bipolar disorder is highly relevant with far-reaching clinical implications. Thus, I believe there is scientific importance to this review topic. I have some feedback regarding the paper's scope, which I describe below.

Overall comments

1) My primary concern with this review is that the issue of delay in the diagnosis and treatment of bipolar disorder is not specific to the 15-24 age range that you define in your protocol but, instead, a problem that cuts across age groups. Many of the studies that you cite also support that this is not an issue specific to adolescents and young adults. The rationale is not clear enough, based on your cited literature and the existing literature in bipolar disorder more generally, for focusing on a young adult and adolescent population in this review (versus a lifespan focus). References 12 and 13 suggest there is a greater treatment delay for younger individuals but this does not detract from the relevance of this topic for those who are older. I think this paper would have greater value if you were to not restrict to adolescents and young adults but, rather, focus this review on the issue of delay in the diagnosis and treatment of bipolar disorder across the lifespan.

• Dear reviewer, thank you for this comment. We agree that better understanding the causes of delay for the diagnosis of bipolar disorder is important. However, for the scope of this work, we decided on a more restricted range for a few reasons. In scoping reviews, feasibility must be balanced with comprehensiveness and breadth. We believe that looking at delays across all ages would be too broad in order to feasibly still conduct a comprehensive study. For example, if we were to not use the adolescent/young adult search criterion, we would be starting with towards 30,000 abstracts to check instead of around 5,000. We further believe that the exclusion of children, and the exclusion of middle-aged and older adults, will help focus our results to be more clear. Childhood-onset bipolar disorder is debated, and may represent a distinct illness, while later-onset bipolar disorder may similarly have different underlying biology or at least be occurring in a very different psychosocial environment where delays are different. Additionally, our age range of adolescence and earlier adulthood does contain the peak of bipolar disorder onset. For this group, developmental stages, involvement of family and location of treatment may have impacts that are not shared with these other ages. 

We have added further discussion of our choice to focus on this age group in the discussion. The relevant paragraph now reads: 

“We believe our focus on those with onset in adolescence and young adulthood will help focus the finding of this work. There is ongoing research into the phenomenological distinction around pre-pubertal-onset BD [https://journalbipolardisorders.springeropen.com/articles/10.1186/s40345-020-00185-2]. Similarly, BD onsetting later in life may have distinct biologic underpinning [https://www.cambridge.org/core/journals/the-british-journal-of-psychiatry/article/elevated-creactive-protein-and-lateonset-bipolar-disorder-in-78-809-individuals-from-the-general-population/4030FC20CB0D9645FCEF7486A69E1338?utm_campaign=shareaholic&utm_medium=copy_link&utm_source=bookmark] and would be occurring in a different psychosocial environment than experienced by youth. Focusing our work on onset in adolescence and young adulthood captures the peak ages of onset and will help determine causes of delay in a population that has long delays and a severe course [https://www.frontiersin.org/articles/10.3389/frcha.2023.1186722/full]. For this group, developmental stages, involvement of family and location of treatment may have impacts that are not shared with these other ages. However, we believe this will allow findings from this scoping review to best help identify the knowledge gaps in the literature and inform clinical practice, as well as future research and policy changes. Services in many settings are structured to specifically address pediatric vs youth vs adult MH, so examining delays specific to this group may better inform targeted interventions at a structural level to improve timely diagnosis and treatment. In addition, we can support recommendations and health promotion for contextual factors relatively unique to this group: families, secondary and tertiary education settings. Investigating delays in this population will also lay a foundation for future work to investigate delays in those experiencing onset in other age groups, and for investigating delays in other psychiatric disorders.” 

2) Related to the above comment, how was the age range of 15-24 decided upon? The cited literature describing age-related delays in treatment of BD does not add clarity here (ie, reference 13 compares < 18 to 18+ - your selected population is represented on both sides of this bracket)

• Thank you for this important comment. We have revised our age range to include individuals with onset between the ages of 13-24. While we acknowledge that the cutoffs are somewhat arbitrary, we believe 13 is an appropriate age as common division between pre-puberty and puberty. Similarly, 25 is often used as a cutoff for transitioning from young adulthood, as this an age by which the majority of adults have started to form a family. https://onlinelibrary.wiley.com/doi/full/10.1111/eip.13181]. This range is also similar to ranges serviced by early intervention services. 

We have added detailed this further in our methods section: 

“In this work, we define adolescence and young adulthood as those experiencing the onset of BD between the ages of 13 and 24. This range is centered upon the mean age of diagnosis, 18 [DSM5], starts with the typical age used as the transition from pre-puberty to puberty, and ends with a common age considered to be the transition from young to middle adulthood as it is around this age that a majority of adults have started to form a family by cohabitation, marriage, or becoming a parent [citations https://www.ncbi.nlm.nih.gov/books/NBK284782/#, https://scholarworks.bgsu.edu/cgi/viewcontent.cgi?article=1009&context=ncfmr_family_profiles]”. This is also a range similar to that used by early intervention services [https://onlinelibrary.wiley.com/doi/pdfdirect/10.1111/eip.13033, https://onlinelibrary.wiley.com/doi/full/10.1111/eip.13181].

3) The information you aim to gather via the Model of Pathways to Treatment is highly specific, and there is a possibility that the research studies you find through your review may not cover all of the questions you seek to answer. How will you handle the limitation of partial information gathered from different studies (ie, a study covering information on some but not all intervals in the Model) and how will the possibility of incomplete information on these intervals within individual studies impact the utility of your narrative summary?

• Thank you for this comment reviewer, this is an excellent point for us to address. We believe that any intervals we do not find relevant studies exploring would itself be informative, as it would motivate future work. We have added this to our limitation section: 

“A limitation of using the Model of Pathways to Treatment framework is its specificity may mean we will not find relevant studies addressing delays in all of its intervals. However, we believe such a finding would itself be informative, as it would motivate future work to investigate a possibly novel source of delay in bipolar diagnosis and treatment. This scoping review will suggest important areas for questioning, and could also be a starting place for consultation/prioritisation activities with stakeholders.” 

Specific line item comments

4) Lines 72-81: The first sentence suggests that the topic of the paragraph covers delays in the diagnosis and treatment of BD among those with pediatric-onset BD but the references here also cover adult onset BD, or cover delays in the diagnosis and treatment of BD more generally. I would recommend re-framing the focus of this paragraph or focusing the paragraph more clearly on your selected population.

• Dear reviewer, thank you for this comment. Citations 12 and 13 specifically reports on how delays in diagnosis are different between those with onset in children and adolescents versus in adulthood, which we are emphasizing in this paragraph. We have deleted the last sentence and that citation which did refer to possible causes of delays in the broad bipolar population. In its place, we have included a sentence including the relevant fact that delays seem to be worse in pediatric-onset BD despite the disease course being more severe:

 This paragraph now reads: 

Delays in diagnosis and treatment tend to be even more prolonged among those with paediatric-onset BD. In an American multisite retrospective study, childhood-onset, adolescent-onset and early adult-onset BDs were associated with 16.8, 11.5 and 4.6 years treatment delays respectively [12]. More recently, data from a Canadian multicenter naturalistic study suggested the median delay of 15 years (IQR: 8.2-25.5) for pediatric-onset BD (<18 years) and 5.0 years (IQR: 1.0-12.0) for adult-onset BD [13]. Delays are longer for pediatric-onset BD despite this onset being associated with a more serious course [Citation Post, R.M. et al, or many others]. 

5) Lines 82-89: Similarly, this paragraph also describes consequences associated with delays in diagnosis and treatment of BD, but upon inspection these papers are not solely in pediatric onset BD but across adult populations of BD. I think the last sentence, "Therefore, there is an urgent need to

88 better understand factors contributing to the delay in the diagnosis and treatment of BD –

89 especially in youth – to inform targeted early identification strategies" is not clearly supported by this paragraph.

• Thank you for this comment, reviewer. We believe that the papers that identify adverse effects of delay in the broader BD population are still relevant, as they can reasonably be assumed to also apply to youth-onset, given that many persons with BD do have onset during this time. As such, we have changed this paragraph to acknowledge that the evidence for these adverse effects come from both pediatric-specific and general BD populations. We have also further detailed in the last sentence why we believe this means investigation of the delays in youth-onset are especially important. 

 The paragraph now reads: 

“Prolonged delays in the diagnosis and subsequent treatment of BD have been shown to have serious consequences for those with pediatric-onset BD, and more broadly in the BD population. Impacts include disruption of crucial age-specific developmental tasks [15], greater severity and frequency of mood episodes [16], higher number of hospitalizations [17], higher number of comorbidities [18,19], and elevated risk of suicide [18,20,21]. In addition, delay in the diagnosis of BD has been shown to be associated with significantly higher healthcare costs [22] as well as higher indirect costs due to work loss [23,24]. Given the more severe course and especially prolonged delay in diagnosis of BD in youth, there is an especially urgent need to better understand factors contributing to the delay in the diagnosis and treatment of BD – to inform targeted early identification strategies. “

Table 1: Eligibility criteria. What was the process for determining age cutoffs (15 to 24)?

• Please see our response to comments 1) and 2) above. We have modified Table 1 for the revised criteria population criteria, which now reads: 

“Adolescents and young adults with onset of mood symptoms or study enrollment at a mean age of 13 to 24 years, with diagnoses of bipolar spectrum disorder (including bipolar I disorder, bipolar II disorder, and bipolar not-otherwise specified, or unspecified/other specified bipolar and related disorder. Studies that include mixed populations whereby more than 80% of participants have bipolar spectrum disorder or where BD specific data can be extracted will also be included.“

Lines 179-183: Please add more clarification on the process of consulting with stakeholders: specifically, how you will use this consultation to add methodological rigor and how you will apply the PLA framework for this review.

• Thank you for this comment. We have changed these lines to be more specific, including the collaboration we will be consulting with, and the specific items we will be consulting on. As this collaboration already incorporates a framework for such interaction, we have removed the specific mention of the PLA framework. 

“As recommended by Levac et al (2010), we will consult with various stakeholders to add to the methodological rigor of the scoping protocol [29]. We will engage with local experts including clinicians, academics as well as individuals with lived experience with BD through the CREST.BD network [https://www.sciencedirect.com/science/article/abs/pii/S0165032715314348] to validate summarization and interpretation of our work, and to inform the future work we call for.”

Lines 192-193: Please add greater explanation on your limitations, specifically the second and third limitations (secondary literature and grey areas).

• Thank you for this comment, we have added greater explanation to these limitations. The relevant paragraph now reads:

“Potential limitations include the exclusion of studies not published in English, secondary research, grey literature, and the focus of this work solely on individuals with onset in adolescence and young adulthood. We excluded secondary studies to avoid redundancy with our own extraction and to ensure our inclusion and exclusion criteria are strictly met, though we acknowledge that findings from these reviews may be relevant. We excluded grey literature to help balance the feasibility of this review with breadth and comprehensiveness [can cite Khalil et al or some of others here], though we acknowledge that this may miss some relevant evidence. This work’s focus on those with bipolar onset in childhood and adolescence does exclude individuals with onset at other points in life. As previously discussed, we believe this will help focus on scope of this work, but it may also exclude important causes for delay found in studies looking at general bipolar populations, and may exclude causes of delay better described for other age-of-onset, but which may still apply to this age group. “

We hope that the above revisions meet with your approval and we look forward to hearing from you.

---

## [Editor Report · Decision Letter 1]

3 Oct 2023

Evidence of factors influencing delays in the diagnosis and treatment of bipolar disorder in adolescents and young adults. Protocol for a systematic scoping review.

PONE-D-23-11500R1

Dear Dr. Keramatian,

We’re pleased to inform you that your manuscript has been judged scientifically suitable for publication and will be formally accepted for publication once it meets all outstanding technical requirements.

Kind regards,

Rikinkumar S. Patel, M.D., M.P.H.

Academic Editor

PLOS ONE

Additional Editor Comments (optional):

Thank you for responding to reviewer's and editorial comments and step-wise fashion, and effectively implementing the required changes. The article looks good for publishing.
---

## [Editor Report · Acceptance letter]

9 Nov 2023

PONE-D-23-11500R1 

Evidence of factors influencing delays in the diagnosis and treatment of bipolar disorder in adolescents and young adults. Protocol for a systematic scoping review. 

Dear Dr. Keramatian:

I'm pleased to inform you that your manuscript has been deemed suitable for publication in PLOS ONE. Congratulations! Your manuscript is now with our production department. 

Kind regards, 

on behalf of

Dr. Rikinkumar S. Patel 

Academic Editor

PLOS ONE